# Reversible interconversion between methanol-diamine and diamide for hydrogen storage based on manganese catalyzed (de)hydrogenation

Zhihui Shao[1], Yang Li[1,2], Chenguang Liu[1], Wenying Ai[1], Shu-Ping Luo[2] & Qiang Liu [1,3]*

The development of cost-effective, sustainable, and efficient catalysts for liquid organic hydrogen carrier systems is a significant goal. However, all the reported liquid organic hydrogen carrier systems relied on the use of precious metal catalysts. Herein, a liquid organic hydrogen carrier system based on non-noble metal catalysis was established. The Mn-catalyzed dehydrogenative coupling of methanol and N,N'-dimethylethylenediamine to form N,N'-(ethane-1,2-diyl)bis(N-methylformamide), and the reverse hydrogenation reaction constitute a hydrogen storage system with a theoretical hydrogen capacity of 5.3 wt%. A rechargeable hydrogen storage could be achieved by a subsequent hydrogenation of the resulting dehydrogenation mixture to regenerate the $H_2$-rich compound. The maximum selectivity for the dehydrogenative amide formation was 97%.

[1] Center of Basic Molecular Science (CBMS), Department of Chemistry, Tsinghua University, Beijing 100084, China. [2] State Key Laboratory Breeding Base of Green Chemistry-Synthesis Technology, Zhejiang University of Technology, Hangzhou 310014, China. [3] School of Biotechnology and Health Sciences, Wuyi University, Jiangmen, Guangdong 529090, China. *email: qiang_liu@mail.tsinghua.edu.cn

Hydrogen has long been regarded as one of the most promising sustainable energy carriers because it has a high mass energy density, can be efficiently transformed into electricity by fuel cells, and generates only water during combustion[1–4]. In recent years, significant progress has been made in the generation of hydrogen from renewable energy sources and the development of efficient hydrogen-powered fuel cells[5–8]. However, hydrogen has not been widely used as an energy source because its storage is challenging. There are economic and safety concerns associated with the compressed and cryogenic liquid hydrogen[9,10], therefore reversible hydrogen storage in chemical bonds via catalytic hydrogenation/dehydrogenation reactions is a promising technique[11,12]. Formic acid[13–16], formaldehyde[17,18], and methanol[19–23] have been extensively studied as molecular hydrogen carriers. However, $H_2$ release from these compounds involves generation of $CO_2$, and $H_2$ cannot be readily reloaded because the liquid carriers are consumed. The need to use stoichiometric amounts of bases and the low hydrogen capacity of formic acid (4.3 wt%) further limit such approaches. To develop more efficient hydrogen storage systems, liquid organic hydrogen carriers (LOHCs) have emerged as a powerful strategy, in which a pair of $H_2$-rich and $H_2$-lean liquid organic compounds can repeatedly release and unload $H_2$[24–28]. Early LOHCs studies focused on the dehydrogenation of cycloalkanes and the reverse hydrogenation of aromatics, but harsh reaction conditions (usually >250 °C) were required[29,30]. To lower the endothermicity, LOHCs systems based on nitrogen-containing organic hydrides[24], e.g., N-heterocycles, were developed. These have high $H_2$ capacities, in the range 5.3–7.3 wt%[31–37]. These systems can be promoted by various homogeneous and heterogeneous catalysts[38–40] under relatively mild conditions (Fig. 1a). Besides, the Milstein and Prakash groups reported LOHCs systems via Ru-catalyzed amide bond formation and hydrogenation (Fig. 1b)[41–45]. Notably, widely available and inexpensive amines and alcohols can be used as hydrogen carriers in these systems. Despite the favorable thermodynamics, usually <80% selectivity for dehydrogenative amide bond formation was reached[43–45].

In addition to identifying renewable and inexpensive liquid molecular hydrogen carriers, the development of cost-effective, sustainable, and efficient catalysts for LOHCs systems is a significant goal. To the best of our knowledge, all the reported LOHCs systems relied on the use of precious metal catalysts. The development of efficient catalytic systems based on earth-abundant non-noble metals is therefore important. On the basis of recent achievements in Mn-catalyzed hydrogenation and dehydrogenation reactions[46–60], we have developed the LOHCs system based on Mn-catalyzed dehydrogenative amide bond formation and the reverse hydrogenation reaction. Remarkably, the maximum selectivity for the dehydrogenative condensation of methanol and N,N′-dimethylethylenediamine (DMEDA) to form N,N′-(ethane-1,2-diyl)bis(N-methylformamide) was 97% (Fig. 1c).

## Results and discussion

**Optimization of the reaction conditions**. We commenced the investigation by exploring the catalytic activities of a series of NNP- and PNP-pincer Mn complexes I to VII in the dehydrogenative condensation of DMEDA (1) and methanol (Table 1). In the presence of PhPNP-complex VI (2 mol%), and tBuOK (4 mol%) in 1,4-dioxane, full conversion of 1 was achieved along with formation of the desired product N,N′-(ethane-1,2-diyl)bis(N-methylformamide) (2a) in 86% yield (Table 1, entry 6). N-methyl-N-[2-(methylamino)ethyl]formamide (2b) and 3-methyl-1-methyleneimidazolidin-1-ium (3) were generated in 9% and 5% yield, respectively. Under the same

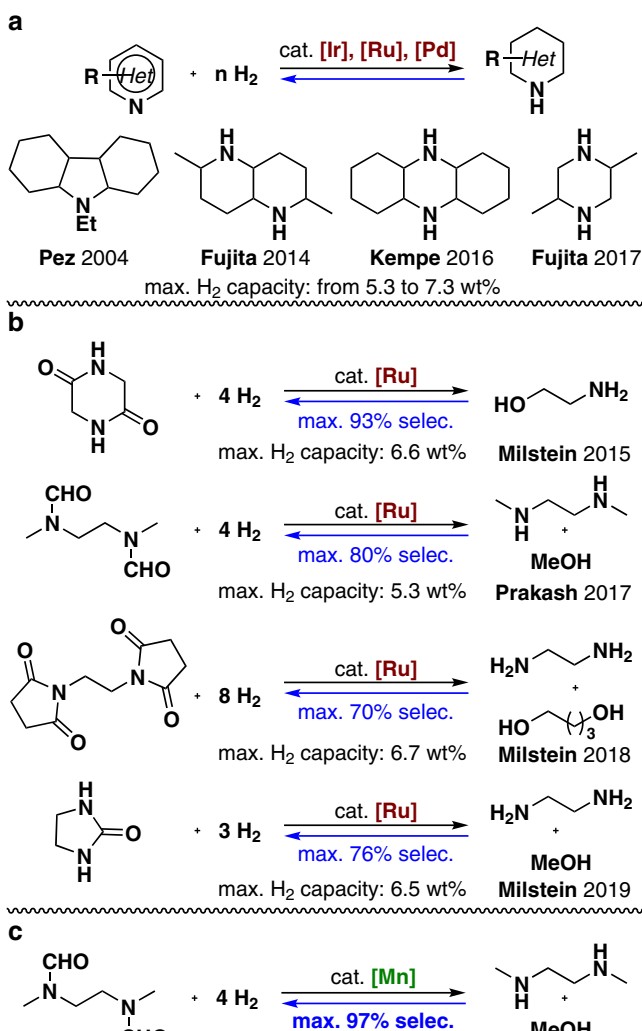

**Fig. 1 Hydrogen storage systems based on N-containing organic hydrides.** **a** Heterocycles hydrogenation and dehydrogenation reactions. **b** Amide bond formation and Hydrogenation reactions. **c** Mn-catalyzed hydrogenation and dehydrogenation reactions.

conditions, the more electron-rich iPrPNP-complex V gave much lower selectivity for 2a (Table 1, entry 5), and the NNP-complexes I–IV showed no activity (Table 1, entries 1–4).

Various reaction parameters were screened to further improve the efficiency of this transformation (Table 2). Use of <6 equiv of MeOH decreased the selectivity for 2a and increased production of monoamide 2b (Table 2, entry 3). The added base also crucially affected the selectivity. The best choice was tBuOK; tBuONa gave much worse results (Table 2, entries 1 and 4). When a weaker base, i.e., KOH, was used, the reaction yield of 2a decreased to 71% (Table 2, entry 6). Concentration screening showed that 0.42–0.63 M 1 gave the highest selectivity for the desired product 2a (Table 2, entries 1 and 8). Decreasing the reaction temperature from 165 to 150 °C diminished the reaction efficiency (Table 2, entry 10). We assumed that $H_2$ generated during the reaction would accumulate in the closed system, and this would inhibit dehydrogenation. To verify this assumption, the evolved gas was released from the reaction system after reaction for 2 h, and the reaction was then performed for a further 6 h at 165 °C. The yield of 2a increased to 97% (Table 2, entry 11). The evolved gas (24 mL) was collected in a gas buret. GC analysis showed that the

## Table 1 Mn-catalyzed dehydrogenative condensation of 1 and CH₃OH.

| Entry | [Mn] | Conv. [%] | 2a [%] | 2b [%] | 3 [%] | H₂ [%][a] |
|-------|------|-----------|--------|--------|-------|-----------|
| 1 | I | 0 | 0 | 0 | 0 | Trace |
| 2 | II | 0 | 0 | 0 | 0 | Trace |
| 3 | III | 0 | 0 | 0 | 0 | Trace |
| 4 | IV | 0 | 0 | 0 | 0 | Trace |
| 5 | V | >99 | 31 | 24 | 45 | 48(>99.9%) |
| 6 | VI | >99 | 86 | 9 | 5 | 92(95.5%) |

Reaction conditions: **1** (0.25 mmol), MeOH (1.5 mmol), **Mn** (2 mol%), *t*BuOK (4 mol%) and 1,4-dioxane (0.4 ml) were reacted at 165 °C for 16 h. The conversion and yield were determined by NMR and GC using 1,1,2,2-tetrachloroethane as the internal standard.
[a]The yield of H₂ was calculated on the basis of maximum H₂ evolution with respect to 100% conversion of **1**–**2a** (4 mmol H₂ per mmol of **1**). The H₂ purity is shown in parentheses.

## Table 2 Optimization of dehydrogenative condensation of 1a and CH₃OH.

| Entry | n | Base | b [mL] | 2a [%] | 2b [%] | 3 [%] | H₂ [%][d] |
|-------|---|------|--------|--------|--------|-------|-----------|
| 1 | 6 | *t*BuOK | 0.4 | 86 | 9 | 5 | 92 (95.5%) |
| 2 | 8 | *t*BuOK | 0.4 | 86 | 5 | 8 | 89 (94.8%) |
| 3 | 4 | *t*BuOK | 0.4 | 80 | 13 | 7 | 85 (96.7%) |
| 4 | 6 | *t*BuONa | 0.4 | 30 | 22 | 46 | 48 (>99.9%) |
| 5 | 6 | KOMe | 0.4 | 85 | 10 | 5 | 90 (98.5%) |
| 6 | 6 | KOH | 0.4 | 71 | 14 | 15 | 80 (99.8%) |
| 7 | 6 | *t*BuOK | 0.2 | 77 | 8 | 15 | 82 (96.4%) |
| 8 | 6 | *t*BuOK | 0.6 | 86 | 11 | 3 | 90 (93.5%) |
| 9 | 6 | *t*BuOK | 1 | 77 | 17 | 5 | 88 (90.2%) |
| 10[a] | 6 | *t*BuOK | 0.4 | 76 | 14 | 7 | 84 (97.8%) |
| 11[b] | 6 | *t*BuOK | 0.4 | 97 | <1 | 2 | 98 (98.7%) |
| 12[c] | 6 | *t*BuOK | 0.4 | 93 | 5 | 2 | 90 (>99.9%) |

Reaction conditions: **1** (0.25 mmol), **VI** (2 mol%), MeOH, *t*BuOK (4 mol%), and dioxane were reacted at 165 °C for 16 h. The conversion and yield were determined by NMR and GC, respectively
[a]The reaction temperature was 150 °C.
[b]After 2 h, the reaction mixture was cooled to room temperature and the evolved gas was released from the system. The temperature was then increased to 165 °C and the reaction was performed for a further 6 h.
[c]**VI** (1 mol%) and *t*BuOK (4 mol%) were used. After 2 h, the reaction mixture was cooled to room temperature and the evolved gas was released from the system. After addition of **VI** (1 mol%), the temperature was increased to 165 °C and the reaction was performed for a further 6 h.
[d]The yield of H₂ was calculated on the basis of maximum H₂ evolution with respect to 100% conversion of **1**–**2a** (4 mmol H₂ per mmol of **1**). The H₂ purity is shown in parentheses.

$H_2$ purity in the gas mixture was 98.7% and the yield was 98%. In addition to $H_2$, 1.3% CO was detected as a result of decarbonylation of the formaldehyde intermediate. The generation of CO was completely suppressed by adding 2 mol% of catalyst **VI** in two equal portions, before and after gas release, after reaction for 2 h. This procedure provided released $H_2$ of purity over 99.9% (Table 2, entry 12).

We then turned our attention to the reverse reaction, hydrogenation of **2a** to diamine **1** (Table 3). The NNP-pincer Mn catalyst **IV** was found to be the most active catalyst for this transformation (Supplementary Table 5). A decrease in the $H_2$

pressure from 60 to 40 bar lowered conversion of the mono-amide **2b** to the fully hydrogenated product **1** at 150 °C (Table 3, entry 2). Notably, this hydrogenation reaction proceeded smoothly at 110 °C, with an excellent yield of **1**, when 2 mol% of catalyst **IV** and 2.5 mol% of *t*BuOK were used (Table 3, entry 8–10). For 0.25 mmol-scale reactions, a larger amount of base was needed to achieve the same level of selectivity (Table 3, entries 1 and 6). The use of a single catalyst for both the dehydrogenation and hydrogenation reactions is desirable toward practical applications. We therefore optimized the hydrogenation of **2a** with catalyst **VI**, the same catalyst as was used in the dehydrogenation reaction (Supplementary Table 8). Under harsh reaction conditions (180 °C, 80 bar of $H_2$), a 94% yield of **1** was obtained with complex **VI** as the catalyst (Table 3, entry 11).

After establishing the optimum reaction conditions for both the dehydrogenative coupling of methanol and **1**, and the hydrogenation of diamide **2a**, we performed Mn-catalyzed reversible interconversion between **1**/methanol and **2a** (Fig. 2). The cycle began with dehydrogenation using 2 mol% **VI** as the catalyst; this resulted in full conversion of **1**. The crude reaction mixture, 2 mol% **IV**, and 2.5 mol% *t*BuOK were transferred to an autoclave for the reverse hydrogenation. The catalytic activity of **IV** was not decreased by the presence of the dehydrogenation catalyst **VI**. After one cycle, diamine **1** was recovered in 95% yield. Moreover, the reversible hydrogenation and dehydrogenation cycle could also be realized using the same catalyst **VI**, albeit with harsh reaction conditions for the $H_2$-loading process. Compared with the corresponding LOHC system based on Ru-catalysis[43], the Mn-catalyzed dehydrogenation process in this system required a higher reaction temperature (165 °C vs 120 °C) and catalyst loading (2 mol% vs 1 mol%), which has yet to be improved in the future works.

**Control experiments**. Under the optimized reaction conditions for catalyst **VI** bearing the N–H moiety, the corresponding *N*-methyl-substituted complex **VII** showed no reactivity for the dehydrogenative condensation of **1** and methanol, and gave only 7% conversion for the hydrogenation of **2a** to **2b**. (Fig. 3) These results demonstrate the crucial role of the N–H group in metal–ligand-cooperation for the (de)hydrogenation process. Furthermore, mercury and ligand poisoning experiments indicated that both hydrogenation and dehydrogenation reaction processes proceed with involvement of homogeneous molecular catalysts (Supplementary Tables 11–13).

Further mechanistic insights into the dehydrogenative condensation reaction were obtained by performing a series of control experiments (Fig. 4). The Mn complex **VI**-catalyzed dehydrogenation of methanol in the absence of diamine **1** only produced methyl formate (**4**) in 4% yield. This shows that dehydrogenative ester formation is unfavorable, therefore aminolysis of **4** with **1** is not the

**Table 3 Optimization of Hydrogenation of 2a.**

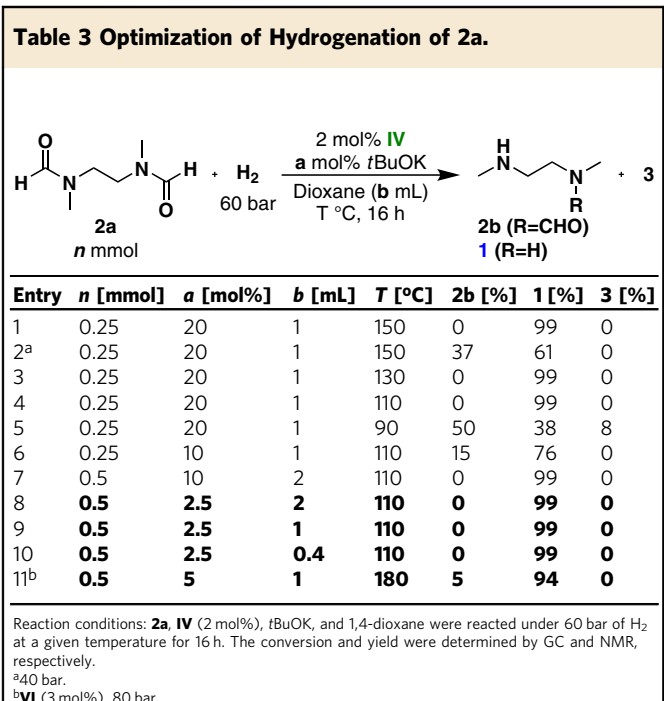

| Entry | $n$ [mmol] | $a$ [mol%] | $b$ [mL] | $T$ [°C] | 2b [%] | 1 [%] | 3 [%] |
|---|---|---|---|---|---|---|---|
| 1 | 0.25 | 20 | 1 | 150 | 0 | 99 | 0 |
| 2[a] | 0.25 | 20 | 1 | 150 | 37 | 61 | 0 |
| 3 | 0.25 | 20 | 1 | 130 | 0 | 99 | 0 |
| 4 | 0.25 | 20 | 1 | 110 | 0 | 99 | 0 |
| 5 | 0.25 | 20 | 1 | 90 | 50 | 38 | 8 |
| 6 | 0.25 | 10 | 1 | 110 | 15 | 76 | 0 |
| 7 | 0.5 | 10 | 2 | 110 | 0 | 99 | 0 |
| 8 | **0.5** | **2.5** | **2** | **110** | **0** | **99** | **0** |
| 9 | **0.5** | **2.5** | **1** | **110** | **0** | **99** | **0** |
| 10 | **0.5** | **2.5** | **0.4** | **110** | **0** | **99** | **0** |
| 11[b] | **0.5** | **5** | **1** | **180** | **5** | **94** | **0** |

Reaction conditions: **2a**, **IV** (2 mol%), *t*BuOK, and 1,4-dioxane were reacted under 60 bar of $H_2$ at a given temperature for 16 h. The conversion and yield were determined by GC and NMR, respectively.
[a]40 bar.
[b]**VI** (3 mol%), 80 bar.

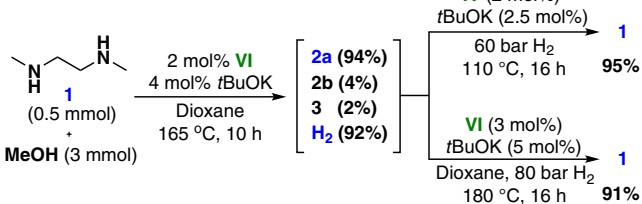

**Fig. 2 Reversible interconversion between 1/methanol and 2a by Mn-catalyzed hydrogenation and dehydrogenation.**

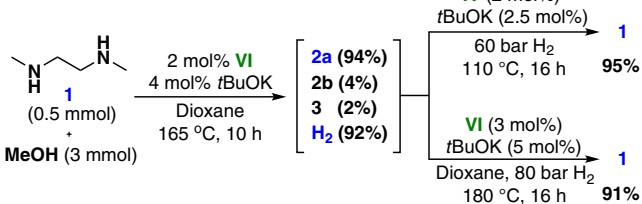

**Fig. 3 Mn-catalyzed hydrogenation and dehydrogenation using N-methyl substituted catalyst VII. a VII**-catalyzed Dehydrogenation reaction. **b VII**-catalyzed hydrogenation reaction. **c** The structure of **VII**.

major reaction pathway for Mn-catalyzed dehydrogenative amide formation (Fig. 4a). Significant amounts of H$_2$ (14.6 mL) and CO (7.4 mL) in a 2:1 ratio were generated in the dehydrogenation of methanol (1.5 mmol). The use of paraformaldehyde instead of methanol led to the formation of **2a**, **2b**, and **3**, albeit with lower selectivities (Fig. 4b). These results indicate that formaldehyde is a possible reaction intermediate in the dehydrogenative condensation of methanol with **1**. Use of monoamide **2b** as the starting material gave **2a** in 92% yield. This shows that **2b** is a likely intermediate in diamide formation (Fig. 4c). In the absence of a

Mn catalyst, **1** reacted with paraformaldehyde to form 3-methyl-1-methyleneimidazolidin-1-ium (**3**) (Fig. 4d). In contrast, **3** was not produced from **2b** under the same conditions (Fig. 4e). This shows that **3** is formed by aminalization of **1** and formaldehyde formed in situ.

Accordingly, to prevent the formation of byproduct **3** in the catalytic dehydrogenative condensation reaction, the Mn-catalyzed dehydrogenation of the hemiaminal intermediate **5** must be much faster than its dehydration under these reaction conditions (Fig. 5).

**Proposed mechanism.** Based on the above results and our previous studies of Mn-catalyzed alcohol dehydrogenation[61–64], plausible reaction pathways for the dehydrogenative condensation of methanol and **1** are proposed (Fig. 6). The dehydrogenation of methanol gives formaldehyde as a reaction intermediate. This reacts with one amine group of **1** to generate the hemiaminal species **5**, which eliminates one molecule of H$_2$ to give the monoamide intermediate **2b**. Reaction of **2b** with another molecule of formaldehyde affords the final product **2a** and one molecule of H$_2$ is released. A possible minor reaction pathway is dehydrogenative ester formation from methanol to form methyl formate (**4**), which undergoes aminolysis with **1** to deliver diamide **2a**. The decarbonylation of formaldehyde to CO and H$_2$ can be prevented in this reaction system by fast condensation of formaldehyde with the amino groups of **1** and monoamide **2b**.

In summary, a LOHC system based on non-noble-metal catalysis was developed. The Mn-catalyzed dehydrogenative condensation of methanol and DMEDA to form N,N'-(ethane-1,2-diyl)bis(N-methylformamide), and the reverse hydrogenation reaction, gave unloading and loading of H$_2$ in excellent yields with a theoretical hydrogen capacity of 5.3 wt%. Reversible hydrogen storage was shown by subsequent hydrogenation of the resulting dehydrogenation mixture to regenerate H$_2$-rich compounds. The maximum selectivity for the dehydrogenation reaction was 97%.

**Fig. 4 Control experiments for mechanistic study. a** **VI**-catalyzed dehydrogenation of methanol. **b** Dehydrogenative condensation of **1** with paraformaldehyde. **c** Dehydrogenative condensation of **2b** with methanol. **d** Condensation of **1** and paraformaldehyde without Mn catalyst. **e** Reactivity of **2b** without Mn catalyst.

**Fig. 5 Analysis of the competitive reaction pathways of intermediate 5.**

## Methods

**General procedure for the dehydrogenation.** All dehydrogenation experiments were carried out in a 25 mL pressure seal tube. In the argon atmosphere glovebox, [**Mn**], tBuOK, solvent, N,N'-dimethylethylenediamine **1**, and MeOH were added sequentially to the seal tube equipped with a magnetic stir bar, The reaction mixture was stirred at given temperature for 16 h and cooled to room temperature. After the gas was released, the conversion of **1** was determined by NMR and the yield of products **2a**, **2b**, and **3** was determined by GC wiith 1,1,2,2-tetra-chloroethane as the internal standard.

**General procedure for the hydrogenation.** All hydrogenation experiments were carried out in a Parr Instruments 4560 series autoclave (300 mL) containing an alloy plate with wells for seven 4 mL glass vials. In the argon atmosphere glovebox, N,N'-(Ethane-1,2-diyl)bis(N-methylformamide) **2a**, [**Mn**], tBuOK, and solvent were added sequentially to the vial equipped with a magnetic stir bar, which was capped with a septum threaded with a syringe. The vial was placed in the alloy plate, which was then placed to the predried autoclave. Once sealed, the autoclave was purged three times with hydrogen, then pressurized to given pressure and

**Fig. 6 Plausible reaction pathways for dehydrogenative condensation of methanol and 1.**

heated at given temperature for 16 h. After reaction, the autoclave was cooled to 0 °C, depressurized. The conversion and yield were determined by GC and NMR with 1,1,2,2-tetrachloroethane as the internal standard.

## Data availability

The authors declare that all the data supporting the findings of this research are available within the article and its supplementary information.

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

## Acknowledgements

We are grateful for the financial supports from the National Natural Science Foundation of China (21822106 and 91845107), the Beijing Municipal Commission of Science and Technology (Z181100005118001), and the Foundation of the Department of Education of Guangdong Province (2018KZDXM070).

## Author contributions

Z.S. and Q.L. designed and performed the experiments. Y.L., C.L., W.A., and S.L. helped to complete the experiments. Q.L. directed the project and wrote the paper. All authors interpreted the results on the paper.

## Competing interests

The authors declare no competing interests.
