## [Peer Review File · Nature Communications]

Reviewers' Comments:

Reviewer #1:

Remarks to the Author:

This is a significant advance in 'greening' the LOHC area and although there are certain limitations of yield and byproduct formation, as the first such example, this deserves publication in the journal.

Reviewer #2:

Remarks to the Author:

This paper describes the new system for hydrogen storage based on the interconversion between N,N'-dimethylethylenediamine and N,N'-(ethane-1,2-diyl)bis(N-methylformamide), achieving a high hydrogen storage capacity of 5.3 wt%. The most important point of this paper would be the employment of non-noble metal manganese instead of usually used noble metals such as ruthenium, iridium, or palladium.

Both release of hydrogen (dehydrogenation) and storage of hydrogen (hydrogenation) were effectively accomplished by the careful optimization of reaction conditions. Analysis of the evolved hydrogen gas and organic byproducts were also soundly done. Mechanistic studies also seem to be very informative. On the basis of above considerations, this referee is positive for the acceptance of this paper in Nature Communications. However, following two important points should be overcome before publication.

1. The catalyst stability is of great importance for the feasibility of its application. The metal catalyst might be readily reduced to metallic particles under high pressure of hydrogen and high temperatures. Therefore, repetitive hydrogen release and storage processes repeatedly, e.g. five to ten cycles, should be conducted.
2. In this system relatively large amount of solvent (dioxane) are employed. From the viewpoint of efficiency as a hydrogen storage system, the use of solvent should be avoided. The authors should conduct the both dehydrogenation and hydrogenation under the conditions without solvent or by using very small amount of solvent. This referee thinks that reactions in large-scale might give better results even in the absence of solvent.

Reviewer #3:

Remarks to the Author:

In this manuscript, the authors report a manganese catalyst that can catalyze amine reforming of methanol as well as the reverse reaction. While the catalyst has been reported previously, its application in reversible methanol reforming has been explored for the first time. It is an important addition to the short list of catalysts which can catalyze both ways efficiently, especially, this is the first non-noble metal catalyst. However, the chemistry of H₂ storage using DMEDA-MeOH discussed in this manuscript is not novel and has already been well reported in 2017. In addition, the Mn catalyst requires much harsher conditions (temperatures of 165-180°C, high catalyst loading of 2 mol%) compared to the existing Ru-catalyzed system (1%). Hence, the economic benefit of using Mn is more than balanced out by the high catalyst and energy input requirements in this system. Nevertheless, this system is still an important study for sustainable chemistry, with scope of improvements.

This manuscript would justify being published in a journal more specific to novel catalyst developments for known chemical process, and not Nature Communications.

The authors must address to the following major issues before resubmitting:

1. Throughout the manuscript, the present work has been compared to the reported Ru-based MeOH-DMEDA system. This has been inaccurately done: (i) the yield of only diamide has been considered, whereas, both diamide and monoamide contribute to H₂ generation. A rational way would be comparing the H₂ yield (which incorporates contribution both species). In this regard, authors' best result of CO-free H₂ gives 90% yield. This is also the best yield reported by Prakash et al. for the Ru-system but with double TON. Also, the conditions of the 2 systems are very different, hence not ideal to be compared ([Ru]:120 oC-1mol% catalyst load vs. [Mn]:165-180oC-2% catalyst load). Authors should rectify accordingly and include correct comparisons in the discussions.
2. This report discusses about H₂ generation/storage. However, most of the reaction schemes do not include H₂ as a product (Table 1, Table 2, Fig. 4 and SI). These should be corrected.
3. H₂ yields should be incorporated in all the dehydrogenation Tables and Figures.

The authors must also address to the following minor issues before resubmitting:

1. H₂ yield in Table 2, Entry 12 does not match with that of Fig. 2. How did the shorter reaction time give more H₂ yield? Please elaborate on this.
2. In the Ru-system, the PNP-iPr showed high efficiency (90% H₂ yield) even better than PNP-Ph. In this paper, the PNP-iPr gives very low yield of the amides (31%). How do the authors explain this behavior?
3. In Table 2, what is the rationale behind adding the catalyst in two portions during course of the reaction (Entry 11 to Entry 12)? Can the authors explain how this diminished CO generation?
4. In Fig. 6, authors should include the pathway to product 3.
5. Is this catalyst active for dehydrogenation of primary amines with MeOH, similar to Milstein's system?
6. Do the authors see any trace urea in the dehydrogenation reaction? In support of the findings, the authors may include a control experiment with the urea, 1,3-dimethyl-2-imidazolidinone.

清华大学

中国北京 100084

Tsinghua University

Beijing 100084, China

COMMENTS TO AUTHOR:

Reviewer: 1

Recommendation: Accept

Comments:

This is a significant advance in 'greening' the LOHC area and although there are certain limitations of yield and byproduct formation, as the first such example, this deserves publication in the journal.

Our response: Thanks for the kind comments from this reviewer. It is worth mentioning that the yield and selectivity of this LOHC system based on Mn-catalysis is competitive or even better than the reported systems based on Ru and Ir systems, although the catalyst loading and reaction temperature

for the dehydrogenation process is slightly elevated.

Reviewer 2: This paper describes the new system for hydrogen storage based on the interconversion between N,N'-dimethylethylenediamine and N,N'-(ethane-1,2-diyl)bis(N-methylformamide), achieving a high hydrogen storage capacity of 5.3 wt%. The most important point of this paper would be the employment of non-noble metal manganese instead of usually used noble metals such as ruthenium, iridium, or palladium.

Both release of hydrogen (dehydrogenation) and storage of hydrogen (hydrogenation) were effectively accomplished by the careful optimization of reaction conditions. Analysis of the evolved hydrogen gas and organic byproducts were also soundly done. Mechanistic studies also seem to be very informative.

On the basis of above considerations, this referee is positive for the acceptance of this paper in Nature Communications. However, following two important points should be overcome before publication.

The catalyst stability is of great importance for the feasibility of its application. The metal catalyst might be readily reduced to metallic particles under high pressure of hydrogen and high temperatures. Therefore, repetitive hydrogen release and storage processes repeatedly, e.g. five to ten cycles, should be conducted.

Our response: Thanks for all the kind comments and questions from this reviewer. Regarding to the concern for the formation of metallic particles as the active catalysts, we performed mercury and ligand poisoning experiments. Different amounts of PMe_3 , PPh_3 and a drop of mercury were added into the dehydrogenation and hydrogenation reaction systems, respectively. No significant inhibitory effect was observed in all these poisoning experiments, which indicated the homogeneous nature of these manganese catalysis system. These new results were added into the supporting information (Supplementary Table 11-13). Meanwhile, a concise discussion on this point was added on the page 10 of the revised manuscript as following: “Mercury and ligand poisoning experiments indicated that both hydrogenation and dehydrogenation reaction processes proceeded with involvement of homogeneous molecular catalysts (Supplementary Table 11-13).”

Supplementary Table 11. Homogeneity test for the dehydrogenation reaction catalyzed by [Mn]-VI.^a

Entry	additive	n [eq]	Conv. [%]	2a [%]	2b [%]	3 [%]
1	--	--	>99	86	9	5
2	Hg	A drop	>99	84	7	8
3	PMe ₃	0.1	>99	86	9	5
4	PMe ₃	0.5	>99	86	9	5
5	PMe ₃	1	>99	85	10	4
6	PPh ₃	0.1	>99	85	10	5
7	PPh ₃	0.5	>99	85	9	5
8	PPh ₃	1	>99	84	9	6

[a] Reaction conditions: **1** (0.25 mmol), MeOH (6 equivi.), **Mn-VI** (0.005 mmol, 2 mol%), *t*BuOK (0.01 mmol, 4 mol%), dioxane (0.4 mL) and phosphines or mercury (equiv. respect to **Mn-VI**) were reacted at 165 °C for 16 h. The conversion and yield were determined by NMR and GC with 1,1,2,2-tetrachloroethane as the internal standard.

Supplementary Table 12. Homogeneity test for the hydrogenation reaction catalyzed by [Mn]-IV.^a

Entry	additive	n [eq]	Conv. [%]	2b [%]	1 [%]	3 [%]
1	--	--	>99	0	>99	0
2	Hg	A drop	>99	0	>99	0
3	PMe ₃	0.1	>99	0	>99	0
4	PMe ₃	0.5	>99	0	>99	0
5	PMe ₃	1	>99	0	>99	0
6	PPh ₃	0.1	>99	0	>99	0
7	PPh ₃	0.5	>99	0	>99	0
8	PPh ₃	1	>99	0	>99	0

[a] Reaction conditions: **2a** (0.5 mmol), **Mn-IV** (0.01 mmol, 2 mol%), *t*BuOK (0.0125 mmol, 2.5 mol%), dioxane (2 mL) and phosphines or mercury (equiv. respect to **Mn-IV**) were reacted at given temperature under given pressure for 16 h. The conversion and yield were determined by GC and NMR with 1,1,2,2-tetrachloroethane as the internal standard.

Supplementary Table 13. Homogeneity test for the hydrogenation reaction catalyzed by [Mn]-VI.^a

[a] Reaction conditions: **2a** (0.5 mmol), **Mn-VI** (0.015 mmol, 3 mol%), *t*BuOK (0.025 mmol, 5 mol%), dioxane (1 mL) and phosphines or mercury (equiv. respect to **Mn-VI**) were reacted at given temperature under given pressure for 16 h. The conversion and yield were determined by GC and NMR with 1,1,2,2-tetrachloroethane as the internal standard.

To further test the stability of the catalyst, we conducted repetitive hydrogen release and storage processes. Unfortunately, the reaction efficiency was decreased after each cycle due to the formation of side product **3**. It illustrated that the stability of the Mn catalyst should be further increased to ensure a fast catalytic dedhydrogenation of the reaction intermediate hemiaminal all the time instead of its dehydration process that leads to the formation of side product **3** (Fig 5). A further evolution of the Mn-catalysts towards this goal is under way in our lab.

[a] The conversion were based on the product of the former step and the yield were based on the amount of **1** before the reaction.

In this system relatively large amount of solvent (dioxane) are employed. From the viewpoint of efficiency as a hydrogen storage system, the use of solvent should be avoided. The authors should conduct the both dehydrogenation and hydrogenation under the conditions without solvent or by using very small amount of solvent. This referee thinks that reactions in large-scale might give better results even in the absence of solvent.

Our response: We totally agree with this reviewer, and tried to reduce the amount of solvent in both dehydrogenation and hydrogenation reactions. For the dehydrogenation reaction on 0.25 mmol scale, the amount of solvent could be lowered to 0.2 mL without obvious decrease of the reaction selectivity (Supplementary Table 4).

Supplementary Table 4. Volume optimization of dehydrogenation solvents. ^a

Entry	c [mL]	Conv. [%]	2a [%]	2b [%]	3 [%]
1	0.4	>99	97	<1	2
2	0.3	>99	90	7	2
3	0.2	>99	84	10	6
4	0.1	90	55	21	10
5	0	61	16	26	14

[a] Reaction conditions: **1** (0.25 mmol), MeOH (6 equiv.), **Mn-VI** (0.005 mmol, 2 mol%), *t*BuOK (0.01 mmol, 4 mol%) and dioxane were reacted at 165 °C. After 2 h, the reaction mixture was cooled to room temperature and the evolved gas was released from the system. The temperature was then increased to 165 °C and the reaction was performed for a further 6 h. The conversion and yield were determined by NMR and GC with 1,1,2,2-tetrachloroethane as the internal standard.

The dehydrogenation reaction on a larger scale (1.5 mmol) was optimized to further reduce the amount of solvent. In the present of 3 equivalent of methanol, **2a** could be obtained in 52% yield along with **2b** in 32% yield under solvent-free conditions.

With respect to the hydrogenation reactions on 0.5 mmol scale, high reaction efficiency could be achieved using at least 0.3 mL of solvent. A further decrease of the amount of solvent will lead to a worse selectivity (Supplementary Table 9-10).

Supplementary Table 9. Volume optimization of hydrogenation solvents catalyzed by [Mn]-IV. ^a

Entry	c [mL]	Conv. [%]	2b [%]	1 [%]	3 [%]
1	2	>99	0	>99	0
2	1	>99	0	>99	0
3	0.8	>99	0	>99	0
4	0.6	>99	0	>99	0
5	0.4	>99	0	>99	0
6	0.3	>99	0	92	5
7	0.2	>99	<1	72	25

[a] Reaction conditions: **2a** (0.5 mmol), **Mn-IV** (0.01 mmol, 2 mol%), *t*BuOK (0.0125 mmol) and dioxane were reacted at 110 °C under 60 bar H₂ for 16 h. The conversion and yield were determined by GC and NMR with 1,1,2,2-tetrachloroethane as the internal standard.

Supplementary Table 10. Volume optimization of hydrogenation solvents catalyzed by [Mn]-VI. ^a

Entry	c [mL]	Conv. [%]	2b [%]	1 [%]	3 [%]
1	1	>99	5	94	0
2	0.8	>99	6	93	0
3	0.6	>99	5	94	0
4	0.4	>99	5	94	0
5	0.3	>99	2	82	14

[a] Reaction conditions: **2a** (0.5 mmol), **Mn-VI** (0.015 mmol, 3 mol%), *t*BuOK (0.025 mmol) and dioxane were reacted at 180 °C under 80 bar H₂ for 16 h. The conversion and yield were determined by GC and NMR with 1,1,2,2-tetrachloroethane as the internal standard.

Reviewer 3: In this manuscript, the authors report a manganese catalyst that can catalyze amine reforming of methanol as well as the reverse reaction. While the catalyst has been reported previously, its application in reversible methanol reforming has been explored for the first time. It is an important

addition to the short list of catalysts which can catalyze both ways efficiently, especially, this is the first non-noble metal catalyst. However, the chemistry of H₂ storage using DMEDA-MeOH discussed in this manuscript is not novel and has already been well reported in 2017. In addition, the Mn catalyst requires much harsher conditions (temperatures of 165-180°C, high catalyst loading of 2 mol%) compared to the existing Ru-catalyzed system (1%). Hence, the economic benefit of using Mn is more than balanced out by the high catalyst and energy input requirements in this system. Nevertheless, this system is still an important study for sustainable chemistry, with scope of improvements. This manuscript would justify being published in a journal more specific to novel catalyst developments for known chemical process, and not *Nature Communications*.

Our response: Thanks for all the comments and questions from this reviewer. To develop more practical hydrogen storage systems, liquid organic hydrogen carriers (LOHCs) has emerged as a powerful strategy. In addition to identifying renewable and inexpensive liquid molecular hydrogen carriers, the development of cost-effective, sustainable, and efficient catalysts for LOHCs systems is a significant goal. Until now, all the reported LOHCs systems relied on the use of precious metal catalysts. In this manuscript, we report **the first LOHCs system based on non-noble catalysis**, using well-defined Mn-catalysts.

Comparing to the pioneering work published by Prakash's group in 2017, in this work we could use Mn-catalyst instead of Ru-catalyst to realize the same DMEDA-MeOH based LOHC system. Although higher reaction temperature (165 °C VS 120 °C) and catalyst loading (2 mol% VS 1 mol%) are required for the dehydrogenation reaction, our system could reach better selectivity for the fully dehydrogenative product diamide (97% VS 80%) and higher H₂ yield (98% VS 90%). Meanwhile, the reaction temperature for hydrogenation reaction in our system could be a little bit lower than that of the reported Ru-system (110 °C VS 120 °C). Therefore, we think the efficiency of this LOHC system based on Mn-catalysis is competitive with the reported Ru-system.

Given the importance of sustainable hydrogen storage and non-noble metal catalysis, we believe that the novelty of our manuscript could reach the high level of *Nature Communications*.

Throughout the manuscript, the present work has been compared to the reported Ru-based MeOH-DMEDA system. This has been inaccurately done: (i) the yield of only diamide has been considered, whereas, both diamide and monoamide contribute to H₂ generation. A rational way would be comparing the H₂ yield (which incorporates contribution both species). In this regard, authors' best result of CO-free H₂ gives 90% yield. This is also the best yield reported by Prakash et al. for the Ru-system but with double TON. Also, the conditions of the 2 systems are very different, hence not

ideal to be compared ([Ru]:120°C-1mol% catalyst load vs. [Mn]:165-180°C-2% catalyst load). Authors should rectify accordingly and include correct comparisons in the discussions.

Our response: Thanks for the kind remind from this reviewer. We agree with this reviewer that it hard to compare these two systems under different reaction conditions, and the H₂ yield is a key factor in term of the practicability of LOHC system. Therefore, we think the higher selectivity for the desired dehydrogenative product diamide is over emphasized in the submitted manuscript. The statements for such a comparison in text are deleted in this revised manuscript. Moreover, we added a description of the limitation of this system in the discussion part of Fig 2 as following: “Compared to the corresponding LOHC system based on Ru-catalysis, the Mn-catalyzed dehydrogenation process in this system required a higher reaction temperature (165 °C VS 120 °C) and catalyst loading (2 mol% VS 1 mol%), which has yet to be improved in the future work.”

This report discusses about H₂ generation/storage. However, most of the reaction schemes do not include H₂ as a product (Table 1, Table 2, Fig. 4 and SI). These should be corrected. H₂ yields should be incorporated in all the dehydrogenation Tables and Figures.

Our response: As the referee suggested, we have repeated all the entries in Table 1, Table 2, Fig. 4 and all the tables of SI part, and measured the H₂ yield and purity for all these reactions. These results were added in the revised manuscript and supporting information, e.g. Table 2 as shown below:

Table 1. Mn-catalyzed Dehydrogenative Condensation of 1 and CH₃OH. ^a

Entry	[Mn]	Conv. [%]	2a [%]	2b [%]	3 [%]	H ₂ [%] ^b
1	I	0	0	0	0	trace
2	II	0	0	0	0	trace
3	III	0	0	0	0	trace
4	IV	0	0	0	0	trace
5	V	>99	31	24	45	48(>99.9%)
6	VI	>99	86	9	5	92(95.5%)

H₂ yield in Table 2, Entry 12 does not match with that of Fig. 2. How did the shorter reaction time

give more H₂ yield? Please elaborate on this.

Our response: The reaction scale of Table 2, Entry 12 is 0.25 mmol, but that of Fig. 2 is 0.5 mmol. Therefore, more H₂ could be produced in a shorter reaction time in the late case.

In the Ru-system, the PNP-*i*Pr showed high efficiency (90% H₂ yield) even better than PNP-Ph. In this paper, the PNP-*i*Pr gives very low yield of the amides (31%). How do the authors explain this behavior?

Our response: To address this issue, some primitive computational investigations were conducted. As shown below, both the PNP-*i*Pr and PNP-Ph based Mn-catalyzed dehydrogenations of MeOH were calculated.

Figure. Free energy profiles for the dehydrogenation of MeOH by Mn-catalysts. The values represent the relative free energy for each intermediates and transition states by PNP-*i*Pr ligand. The values in parentheses represent the relative free energy by PNP-Ph ligand.

The calculated results show that the Mn-OMe intermediate is the resting state, and the H₂ release is rate-determining step among the catalytic cycle. The overall activation free energy for the

dehydrogenation of MeOH by PNP-*i*Pr-Mn and PNP-Ph-Mn are 29.9 and 36.0 kcal/mol, respectively. The high overall activation free energy for PNP-*i*Pr-Mn system is mainly attributed to the higher thermal stability of Mn-OMe intermediate. The predicted reactivity well correlates the observed yield. A comprehensive mechanistic study to further differentiate and evaluate the reactivity of Mn pincer catalysts is ongoing in our group.

In Table 2, what is the rationale behind adding the catalyst in two portions during course of the reaction (Entry 11 to Entry 12)? Can the authors explain how this diminished CO generation?

Our response: We added the catalyst in two portions to reduce the dehydrogenation rate of methanol to ensure a low concentration of the initial dehydrogenation product formaldehyde in the reaction system. In this case, the produced formaldehyde could react with amines in excess amounts very fast to avoid its decarbonylation to produce CO gas as the side product.

In Fig. 6, authors should include the pathway to product 3.

Our response: As the reviewer suggested, we added the reaction pathway to product 3 in Fig. 6 of the revised manuscript.

Fig 6. Plausible reaction pathways for dehydrogenative condensation of methanol and 1.

Is this catalyst active for dehydrogenation of primary amines with MeOH, similar to Milstein's system?

Our response: We are working on the Mn-catalyzed dehydrogenative condensation of primary amine with MeOH. At the moment, high conversion and moderate selectivity have been achieved. It well demonstrates the feasibility of using Mn-catalyst to realize this transformation. A further optimization of this system is underway in our lab.

Entry	Mn	Conv. [%]	B [%]	C [%]	D [%]
1	V	>99	60	5	22
2	VI	>99	31	17	26

Do the authors see any trace urea in the dehydrogenation reaction? In support of the findings, the authors may include a control experiment with the urea, 1,3-dimethyl-2-imidazolidinone.

Our response: We didn't detect any urea product in the dehydrogenation reaction. Furthermore, we performed control experiments for both dehydrogenation and hydrogenation reactions using 1,3-dimethyl-2-imidazolidinone as the substrate. No conversion was observed in both reactions, which indicated that urea was not the reaction intermediate for these reactions.

Thank you very much for your time and consideration, and we hope you would find the revised manuscript acceptable. We certainly look forward to hearing from you regarding the progress of processing this manuscript.

With best regards

Qiang Liu

Reviewers' Comments:

Reviewer #2:

Remarks to the Author:

The author has improved the manuscript appropriately as requested.

I think that the revised version is now acceptable for publication in Nature Communications.

Reviewer #3:

Remarks to the Author:

The comments of this reviewer have been well addressed by the authors. They have addressed most of the points in great details by performing additional experiments, repeating most of the experiments and also performing some computational studies. The manuscript in its current form appears to be well-written and carefully edited. I recommend it to be published in Nature Communications as is.

清华大学
中国北京 100084

Tsinghua University
Beijing 100084, China

Dr. Qiang Liu, *Associate Professor*
Chemistry Department, Tsinghua University
Beijing 100084, P. R. China
qiang_liu@mail.tsinghua.edu.cn
<http://www.researcherid.com/rid/A-8654-2014>

November 22th, 2019

COMMENTS TO AUTHOR:

Reviewer: 2

Recommendation: Accept

Comments:

The author has improved the manuscript appropriately as requested.

I think that the revised version is now acceptable for publication in *Nature Communications*.

Our response: We appreciate your recommendation of acceptance and helpful comments in the reviewing process and are pleased to have our manuscript be reviewed by you.

Reviewer: 3

Recommendation: Accept

Comments:

The comments of this reviewer have been well addressed by the authors. They have addressed most of the points in great details by performing additional experiments, repeating most of the experiments and also performing some computational studies. The manuscript in its current form appears to be well-written and carefully edited. I recommend it to be published in *Nature Communications* as is.

Our response: Thank you very much for your valuable comments in the reviewing process, which have helped us improve the quality of the whole manuscript. We sincerely appreciate you for recommending our manuscript be accepted by *Nature Communications*.

Thank you very much for your time and consideration.

With best regards

Qiang Liu